# Finite Element Analysis of Crack Propagation in Precise Separation Process of Tubing Eccentric Loading Fatigue Fracture

Renfeng Zhao *, Runze Pan, Weicheng Gao, Dongya Zhang, Xudong Xiao, Pengkang Zhao and Xiaohuan Zhu

Department of Mechanical and Precision Instrument Engineering, Xi'an University of Technology, Xi'an 710048, China; 2210221268@stu.xaut.edu.cn (R.P.); 2200220034@stu.xaut.edu.cn (W.G.); dyzhang@xaut.edu.cn (D.Z.); xiaoxd@xaut.edu.cn (X.X.); zhaopengkang@xaut.edu.cn (P.Z.); 3180211067@stu.xaut.edu.cn (X.Z.)
* Correspondence: zrf20070607@xaut.edu.cn

**Abstract:** Aiming at the difficulties of efficient and precise separation of metal tubes. For tubing fatigue fracture precision separation, this paper proposes a tubing precision separation process under eccentric wheel rotational bending fatigue loading. Mechanical properties experiments for 304 stainless steel tubing are carried out. On this basis, the J-C constitutive model of 304 stainless steel tubing fatigue fracture is established. In addition, the tubing stress at different rotation angles of the eccentric wheel and the axial stress at four positions of the tubing V-groove section are analyzed, and the process of the tubing precision separation is simulated. According to the results, the axial stress of the tubing V-groove section changes basically symmetrically, and the stress is the largest at the smallest distance from the eccentric wheel excircle axis. During fatigue loading cycles from 0 to 500, the crack growth rate of tubing outer ring is greater than that of the inner ring. With the increase in loading cycles, the crack surface gradually changes from smooth to undulating. Finally, an obvious final fracture region appears, which is consistent with the experimental results.

**Keywords:** tubing; crack propagation; J-C constitutive model; finite element simulation

## 1. Introduction

With the continuous development of aviation, aerospace, energy, automobile, electronic products and biomedical engineering, the key components of high-end equipment have the characteristics of light weight, high performance and great efficiency. As an important key component, hollow shaft parts are widely used in automobiles, high-speed rail and aerospace equipment [1–6]. Tubing separation is the first step of tubing parts processing, which directly affects the quality and efficiency of subsequent processing. The original blanks, such as hollow pins, outer rings in bearings, bushings for chains and retaining rings, are obtained through tubing blanking [7–9].

There is a direct link between the quality of tubing separation and the quality and efficiency of subsequent processing. The existing tubing separation methods are mainly divided into the cutting method and shearing method, most of which have problems, such as low processing efficiency, poor section accuracy and serious waste of raw materials. The cutting method mainly includes sawing, turning and laser cutting. The shearing method mainly includes the shearing method without mandrel, the shearing method with mandrel and the circumferential shearing method with mandrel [10,11]. The tubing sections obtained by sawing and shearing separation generally have shortcomings, such as low separation efficiency, large saw loss, noise and long burrs [12,13]. Due to the hollow structure, metal tubing is prone to deformation when subjected to radial impact or pressure, which seriously affects the blanking quality. This is also considered a major difficulty in tubing separation. New processes and blanking methods are necessary to obtain acceptable section quality. Therefore, tubing precision separation technology with high quality, high

efficiency and low energy consumption has become one of the key research directions in the field of machinery manufacturing [14–16].

As for the problems of poor section precision and serious waste of raw materials in tubing blanking, this paper adopts the tubing fracture separation method of eccentric wheel rotational fatigue loading, which is a crack-controlled propagation precision separation technology. By prefabricating annular notch on specific tubing sections and applying cyclic loading, the tubing will generate fatigue cracks at the notch and expand along the specific section until the purpose of tubing separation is achieved. Compared with the traditional tubing separation method, this method has the advantages of simple structure, small loading stress, good separation efficiency, high section quality, as well as saving raw materials. In order to study the tubing fatigue fracture under eccentric wheel loading, the extended finite element method is used to simulate and analyze the tubing fatigue crack propagation. In addition, the key issues and fracture mechanism involved in the tubing separation process under eccentric wheel rotational fatigue loading were studied deeply, so as to better realize the controlled precision separation of tubing. Today's development concepts of green manufacturing and efficient and precise forming urgently require new efficient and precise separation methods for tubes to provide technical support. Exploring new methods of separating metal tubes with high efficiency, precision and material saving has important engineering application value. In studying the performance of tubes under complex fatigue, the theory of crack propagation under load has important scientific significance.

## 2. Principle of Tubing Precision Separation under Eccentric Wheel Rotational Bending Fatigue Loading

This efficient precision separation method comprehensively utilizes the rapid initiation of micro-cracks and crack propagation technology, first artificially prefabricating the annular grooves on the surface of the tube. Through the high-speed rotation of the specially designed eccentric wheel, a periodic cyclic bending fatigue load is applied to the radial direction of the tube. Under the combined action of the stress concentration effect of the annular groove and the dynamic bending fatigue load, the root of the annular groove of the tube quickly initiates micro-cracks and causes macroscopic cracks. In the section of the annular groove of the tube, it continuously expands to the center of the tube according to the specified direction and expansion rate and finally realizes the efficient, precise and controllable separation of the tube. The principle is shown in Figure 1.

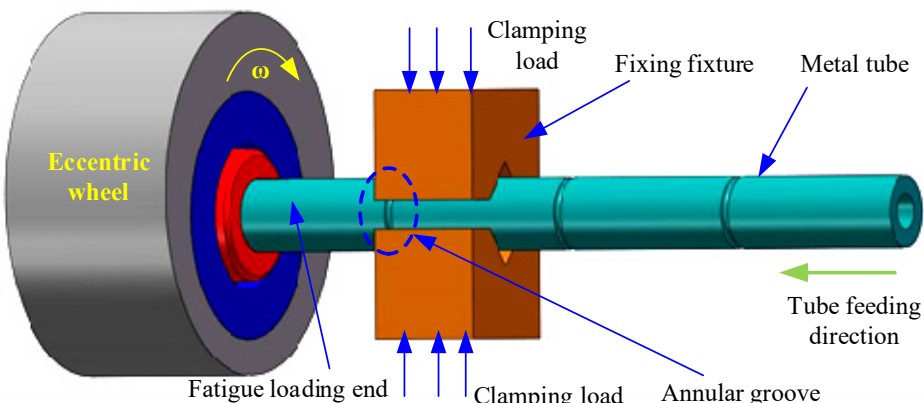

**Figure 1.** Schematic diagram of tubing eccentric wheel rotational bending fatigue loading.

## 3. Metal Tubing Mechanical Properties Experiment

### 3.1. Experimental Tubing

In finite element analysis, material properties have a significant impact on experimental results. With different production processes, the mechanical properties of tubing and bars are significantly different. In order to accurately obtain the mechanical properties of

tubing, this paper uses tensile experiment to explore the mechanical properties of a metal tubing section sample, which can be used for subsequent finite element analysis research.

With the advantages of high strength, high-temperature resistance and corrosion resistance, 304 stainless steel is widely used in various industries. Therefore, 304 stainless steel is selected as the experimental object. The specific chemical composition is shown in Table 1.

**Table 1.** Chemical composition of 304 stainless steel material.

| Element | C | Si | Mn | P | S | Ni | Cr |
|---|---|---|---|---|---|---|---|
| Content/% | 0.07 | 0.74 | 1.50 | 0.035 | 0.03 | 7.85 | 18.53 |

### 3.2. Design and Installation of Tensile Experiment

The tensile experiment refers to the national standard "GB/T228.1-2010" [17] for the tensile experiment method at room temperature of metal materials. When the tubing outer diameter $D$ is less than 30 mm, the tensile sample adopts the section sample. The tubing outer diameter in this experiment is set to 20 mm, and the obtained geometrical dimensions of the tensile sample are shown in Figure 2. In more detail, the sample outer diameter $D$ is 20 mm, the inner diameter $d$ is 14 mm, the gauge length $L_0$ is 72 mm, the parallel length $L_c$ is 90 mm, and the sample total length $L$ is 190 mm. In order to prevent the section sample from being flattened, both ends of the sample are equipped with stoppers. In order to ensure reliability, the clamping parts at both sample ends are machined with grooves. The section sample and stopper real sample are shown in Figure 3.

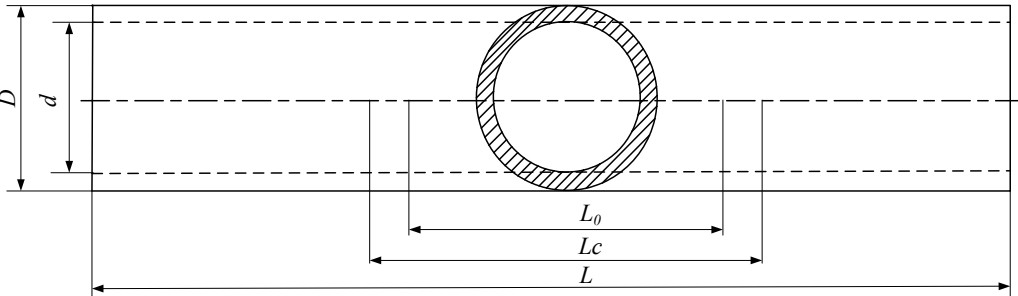

**Figure 2.** Schematic diagram of tensile experiment section sample.

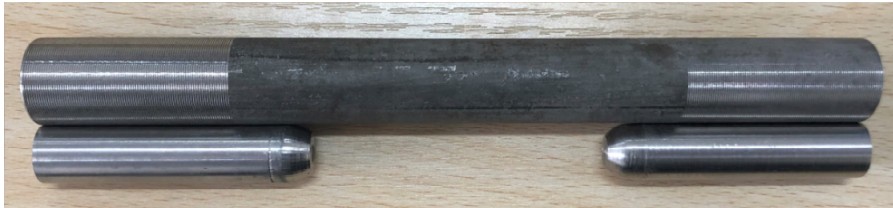

**Figure 3.** Section sample and stopper.

Considering that the deformation temperature in the experiment is 25 °C, tensile experiments with strain rates of 0.0001, 0.0005, 0.001 and 0.01 s$^{-1}$ are chosen.

CMT-300 electronic universal experimental machine is used for tensile experiment, with a maximum load of 300 kN. The automatic signal acquisition instrument with CMT-300 material testing machine is used to collect the tensile force. The longitudinal extensometer is used to obtain axial displacement. The extensometer model is YYU10/50, with a gauge length of 50 mm and a range of 25 mm.

### 3.3. Tensile Experiment Results and Analysis

The tensile results and section of 304 stainless steel section sample are shown in Figure 4. It can be seen that the sample is pulled off in the middle section, and the breakage is obviously necked and fractured in the direction of 45°. The true stress–true strain curve on quasi-static tensile experiment of 304 stainless steel section sample at room temperature is shown in Figure 5. At room temperature, the yield strength and tensile strength of 304 stainless steel gradually increase with the increase in strain rate.

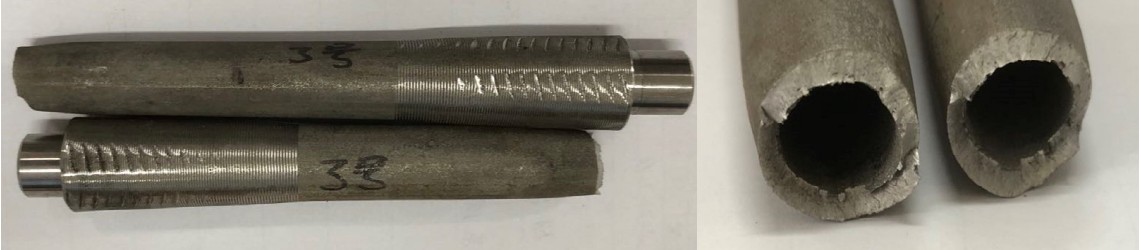

**Figure 4.** Tensile results of section sample and section.

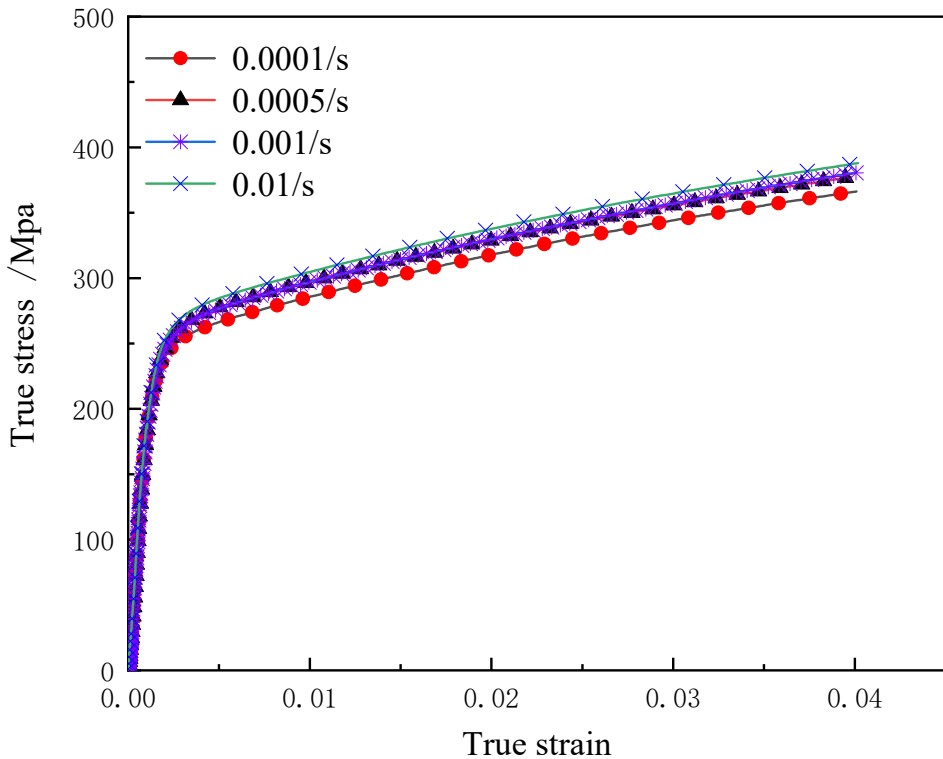

**Figure 5.** 304 stainless steel tubing true stress–true strain curve.

### 3.4. Establishment of Constitutive Model

Due to its simple form, few measured parameters and clear meaning, the Johnson–Cook model can well reflect the influence of strain, strain rate and temperature on the mechanical properties of ductile metal materials during the deformation process, which is widely used in the study of mechanical properties of materials [18]. The Johnson–Cook model expression is as follows:

$$\sigma = (A + B\varepsilon^n)\left(1 + C \ln \dot{\varepsilon}^*\right)\left(1 - T_*^m\right) \tag{1}$$

where $A$ is the yield strength of the material at the reference temperature and reference strain rate; $B$ and $n$ are the strain hardening coefficients; $C$ is the strain rate sensitivity

coefficient; $m$ is the temperature softening index; $\varepsilon$ is the equivalent plastic strain; $\dot{\varepsilon}^*$ is the dimensionless equivalent plastic strain rate. In $\dot{\varepsilon}^* = \dot{\varepsilon}/\dot{\varepsilon}_0$, $\dot{\varepsilon}$ is the actual strain rate, and $\dot{\varepsilon}_0$ is the reference strain rate. In $T_* = (T - T_r)(T_m - T_r)$, $T_r$ is the reference temperature, $T_m$ is the melting point of the material, and $T$ is the experimental temperature. Due to the small effect of temperature during the tubing fatigue precision fracture separation process, this tensile experiment is carried out at room temperature. Only four parameters, $A$, $B$, $n$ and $C$, need to be obtained.

(1)    Determine the parameters $A$, $B$ and $n$.

Since the experiment is carried out at room temperature, both the experimental temperature and the reference temperature are 25 °C. $\dot{\varepsilon} = 0.001$ s$^{-1}$ Therefore, Equation (1) can be simplified as follows:

$$\sigma = (A + B\varepsilon^n)\left(1 + C\ln\frac{\dot{\varepsilon}}{0.001}\right) \tag{2}$$

When the strain rate $\dot{\varepsilon} = 0.001$ s$^{-1}$, Equation (2) can be simplified as follows:

$$\sigma = A + B\varepsilon^n \tag{3}$$

At this point, the yield strength of the material under stretch $\dot{\varepsilon} = 0.001$ s$^{-1}$ is the value of $A$. According to Figure 5, $A$ = 245 MPa can be obtained. By taking the logarithm of both sides, Equation (3) can be transformed into the following:

$$\ln(\sigma - A) = \ln B + n\ln\varepsilon \tag{4}$$

By substituting the obtained true stress–true strain data into the above equation, $n$ = 0.35 and $B$ = 460 MPa.

(2)    Determine the parameter $C$.

At room temperature, when the plastic strain $\varepsilon$ = 0, Equation (2) can be simplified as follows:

$$\sigma = A\left(1 + C\ln\dot{\varepsilon}^*\right) \tag{5}$$

According to the tensile experiment data, the parameter $C$ under different strain rates can be obtained. Through nonlinear fitting, the parameter $C$ = 0.0038 can be obtained.

Therefore, the relationship between true stress and true strain of 304 stainless steel tubing at room temperature can be expressed as follows:

$$\sigma = \left(245 + 460\varepsilon^{0.35}\right)\left(1 + 0.0038\ln\frac{\dot{\varepsilon}}{0.001}\right) \tag{6}$$

Through tensile experiment, the Johnson–Cook constitutive model parameters of 304 stainless steel tubing obtained are shown in Table 2.

**Table 2.** Johnson–Cook model parameters of 304 stainless steel.

| Material | Reference Strain Rate | $A$ (MPa) | $B$ (MPa) | $n$ | $C$ |
|---|---|---|---|---|---|
| 304 stainless steel | 0.001 s$^{-1}$ | 245 | 460 | 0.35 | 0.0038 |

## 4. Analysis of Tubing Stress Changes under Eccentric Wheel Fatigue Loading

### 4.1. Establishment of Eccentric Wheel Loading Finite Element Model

By the extended finite element method (XFEM), the crack propagation process on the tube surface is simulated. Under eccentric wheel rotational fatigue loading, the finite element simulation model of tubing force analysis consists of two parts, which are tubing and eccentric wheel. The tubing geometry is shown in Figure 6. In more detail, the outer

diameter $D$ of 304 stainless steel tubing is 20 mm, the inner diameter $d$ is 14 mm, and the total length $L$ is 120 mm. The left end is the fixed end, with a length $L_1$ of 20 mm. The right end is the loading end, with a length $L_2$ of 100 mm. The tubing surface annular V-notch flare angle $\alpha$ is 90°, the notch depth $h$ is 0.8 mm, and the notch base angle radius $r$ is 0.2 mm. The geometric dimensions of eccentric wheel are shown in Figure 7. The excircle diameter of eccentric wheel is 34 mm, the inter-circle diameter is 20 mm, the height is 5 mm, the inter-circle chamfer is 2 mm, the angle is 45°, and the eccentricity of excircle axis $O_1$ and inter-circle axis $O_2$ is 3 mm.

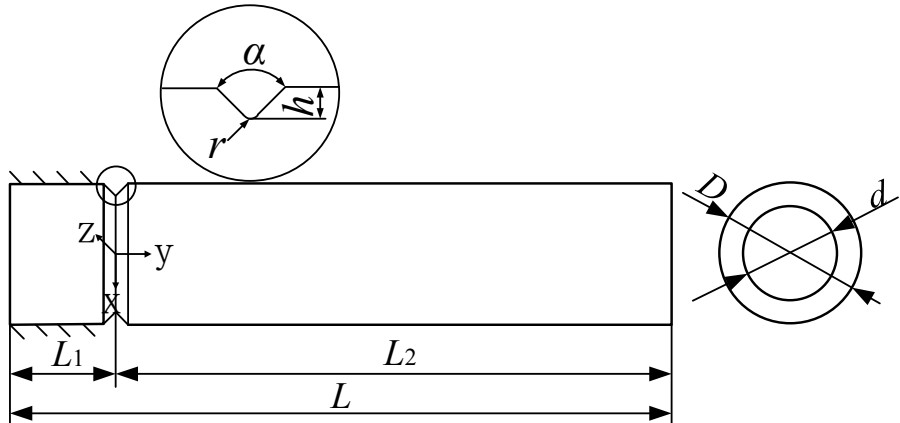

**Figure 6.** Tubing geometry parameters.

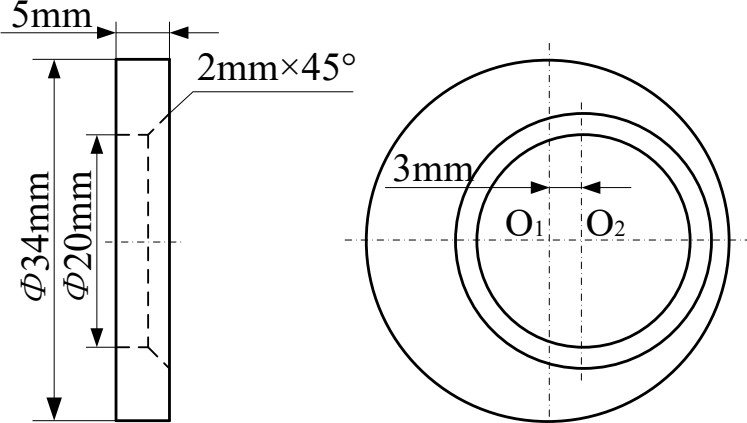

**Figure 7.** Eccentric wheel geometry.

The finite element model assemblies of tubing and eccentric wheel are shown in Figure 8. During the experiment, the left end of the tubing is fixed. The eccentric wheel rotates clockwise around the excircle axis $O_1$ and applies cyclic loading to the tubing through the inter-circle. The meshing of tubing finite element model uses linear hexahedral element with eight nodes. The total number of nodes is 57,288, and the total number of cells is 48,732. In the reduced integral element calculation, the element size of the main tubing part is 1 mm. The V-groove notch part is processed with local mesh refinement, with a cell size of 0.5 mm. Its material parameters are the 304 stainless steel tubing J-C constitutive model requested above. Since the eccentric wheel is a discrete rigid body, the meshing also adopts a linear hexahedral element with eight nodes.

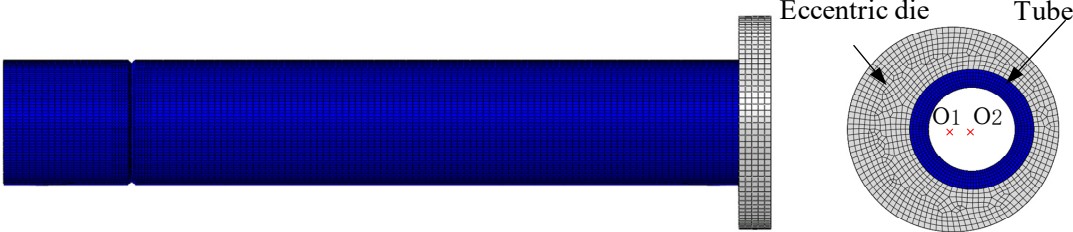

**Figure 8.** Assembly of tubing and eccentric wheel.

### 4.2. Stress Variation of Tubing under Different Rotation Angles of Eccentric Wheel

Under eccentric wheel loading, the axial stress nephograms of tubing V-groove section at different rotation angles are shown in Figure 9. When the rotation angle of the eccentric wheel is between 0° and 90°, the stress concentration position of tubing V-groove section changes continuously, and the maximum axial tensile stress and compressive stress increase continuously. During the expansion process, the axial stress gradually expands from the V-groove outer ring to the inner ring. When the rotation angle of eccentric wheel is 135°–225°, the value of the maximum axial tensile stress does not change much, but the axial stress of the inner ring in tubing V-groove section increases continuously. The axial compressive stress of the tubing V-groove section is greatest when the eccentric wheel is rotated to 180°. At this time, the force-bearing end reaches the maximum eccentricity, with a displacement value of 6 mm. When the rotation angle of the eccentric wheel is 270°–360°, the axial stress of the tubing V-groove section is greater than that from 0° to 225°. As the eccentric wheel rotates, the direction of force on the tubing changes. The shift value of the loading end position under different rotation angles is different. The axial tensile stress of the tubing V-groove section is greatest when the eccentric wheel is rotated to 360°. According to the stress contour, the axial maximum tensile stress of the tubing V-groove section is at the same position as the maximum compressive stress. The tube after compression will have a work hardening effect, which will reduce the plasticity. When the tubing returns to its original state, the local tensile stress becomes larger.

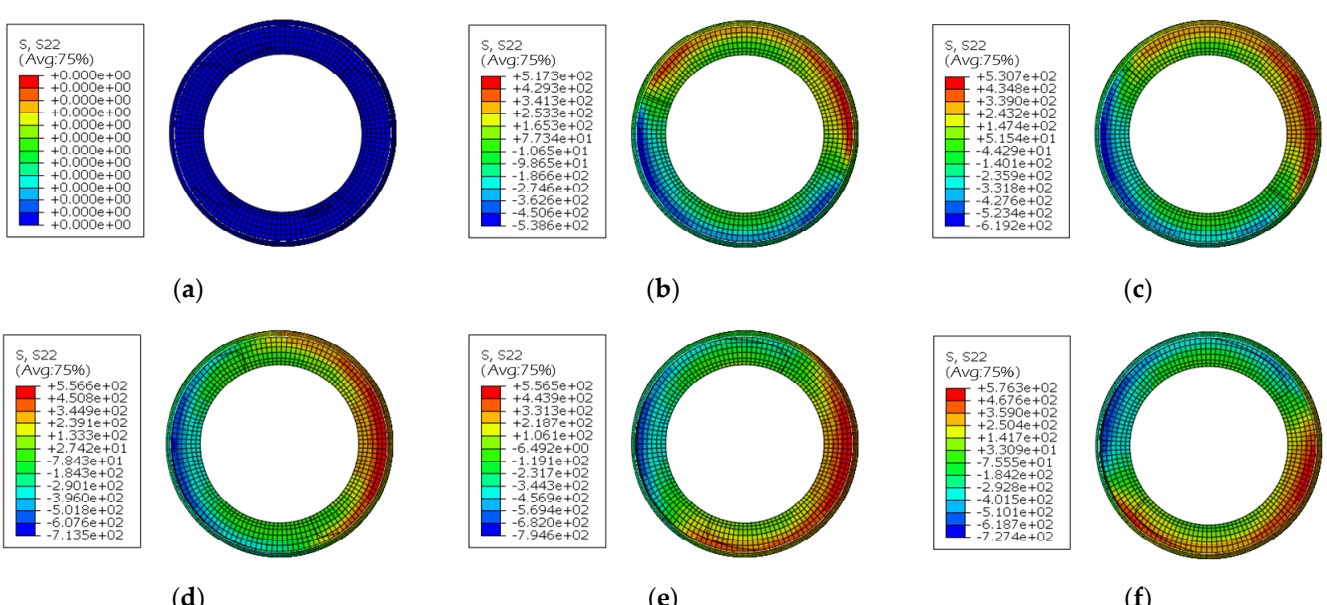

**Figure 9.** *Cont.*

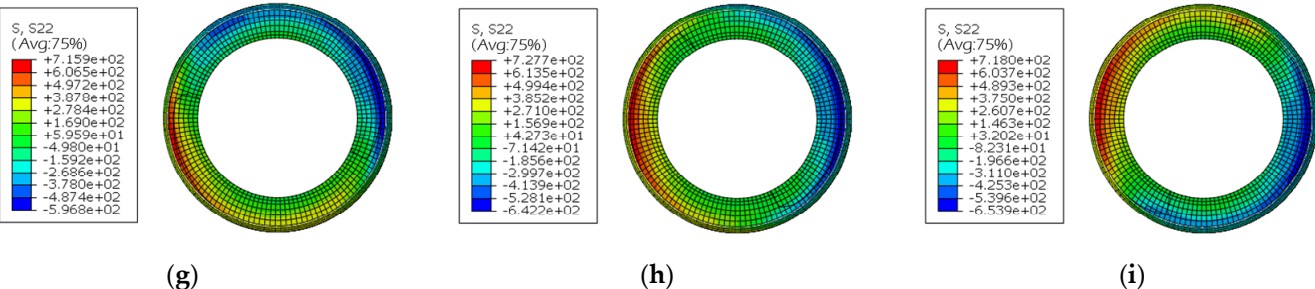

**Figure 9.** Contour image of axial stress of tubing V-groove section under eccentric wheel loading: (**a**) Eccentric angle = 0°; (**b**) Eccentric angle = 45°; (**c**) Eccentric angle = 90°; (**d**) Eccentric angle = 135°; (**e**) Eccentric angle = 180°; (**f**) Eccentric angle = 225°; (**g**) Eccentric angle = 270°; (**h**) Eccentric angle = 315°; (**i**) Eccentric angle = 360°.

### 4.3. Axial Stress Variation of Tubing V-Groove Section at Different Positions during Fatigue Loading

In order to further study the variation law of tubing stress under eccentric wheel loading, four different points (point a, point b, point c and point d) near the outer surface of the tubing V-groove section are analyzed, as shown in Figure 10. Point b is the farthest from the eccentric wheel excircle axis, and point d is the closest to the eccentric wheel excircle axis.

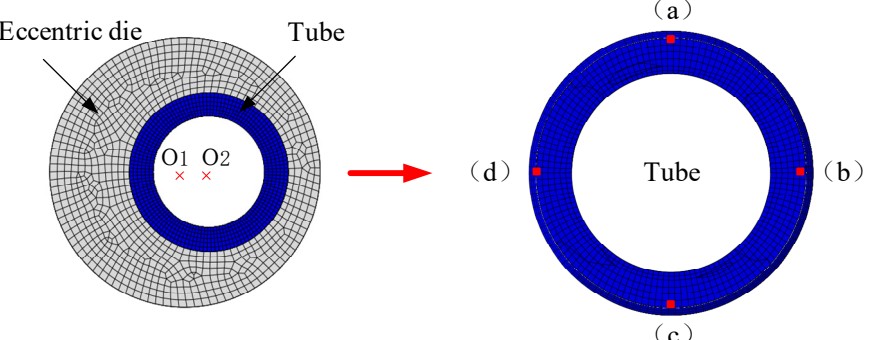

**Figure 10.** Sampling locations for stress analysis of tubing V-groove section.

The variation of axial stress at different positions in the tubing V-groove section under eccentric wheel loading is shown in Figure 11. Under eccentric wheel cyclic loading, different locations in the tubing V-groove section are subjected to tensile stress and compressive stress. The magnitude and time of tensile stress and compressive stress at different locations are different.

It can be seen from Figure 11a that at 0–0.36 s, the tensile stress on point a is the same as the compressive stress on point c. The tensile stress and compressive stress of point a and point c are both less than 400 MPa. At 0.36 s, the axial stress of point a and point c is 0 MPa. At 0.36–1 s, point a and point c change symmetrically because they have the same eccentricity relative to the eccentric die. In addition, point a and point c in the V-groove section are in the symmetrical position of the tubing, and their stress changes are symmetrical to each other.

It can be seen from Figure 11b that at 0–0.62 s, the duration of tensile stress on point b is the same as the duration of compressive stress on point d. At 0.62 s, the axial stress of point b and point d is 0 MPa. The maximum compressive stress on point b is greater than the maximum tensile stress. In more detail, the maximum compressive stress is greater than 650 MPa, and the maximum tensile stress is less than 600 MPa. The compressive stress duration is less than the tensile stress duration. The maximum compressive stress on point d is greater than the maximum tensile stress. In more detail, the maximum compressive stress is greater than 750 MPa, and the maximum tensile stress is less than 700 MPa. The

duration of compressive stress is greater than that of tensile stress. The eccentricity of point b and point d at different positions of the tube is different. According to the work hardening effect, the greater the eccentricity, the greater the axial stress. The minimum tensile stress and compressive stress are at point a and point c. Point d closest to the eccentric wheel excircle axis has the largest compressive stress.

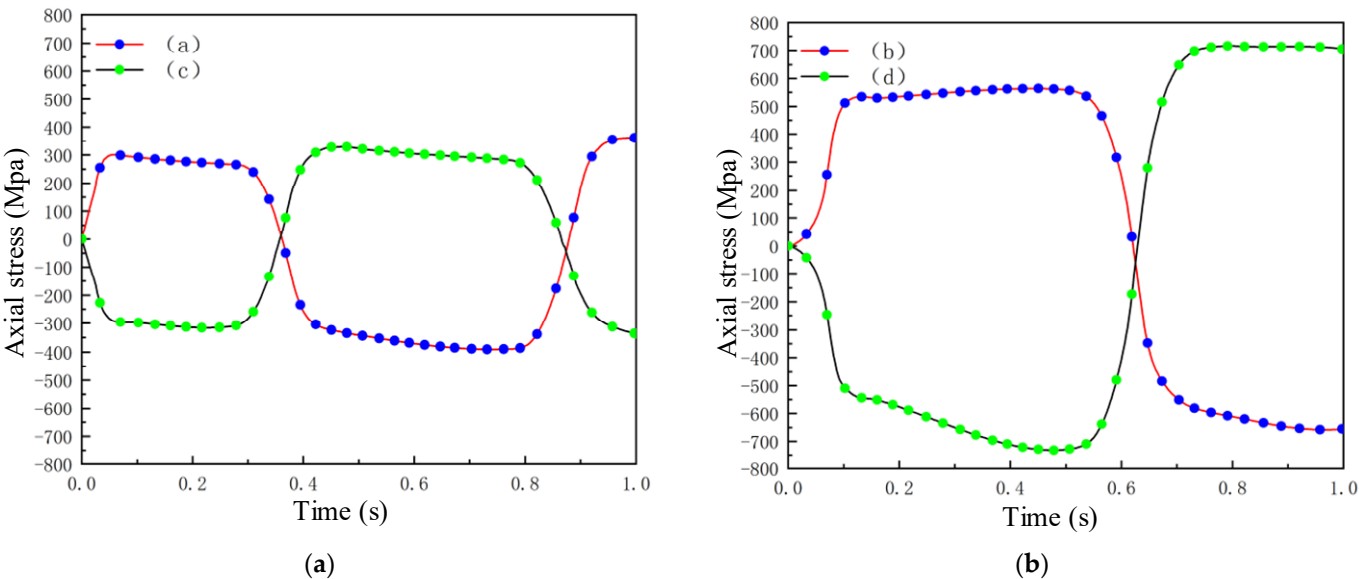

(**a**)　　　　　　　　　　　　　　　　　　(**b**)

**Figure 11.** Variation of axial stress at four positions of V-groove section under eccentric wheel loading: (**a**) Variation of axial stress at point a and point c; (**b**) Variation of axial stress at point b and point d.

## 5. Tubing Fatigue Precision Fracture Separation Finite Element Simulation

### 5.1. Tubing Fatigue Fracture Precision Separation Finite Element Model

Extended finite element method is a new numerical analysis method to study crack propagation. With the advantages of conventional finite element method (CFEM), it can deal with various discontinuities, such as crack growth and complex fluids, and it is widely used in the field of material failure. Through ABAQUS finite element simulation software, the tubing fatigue fracture finite element model is established. In addition, XFEM is used to analyze the fatigue crack propagation process.

The established tubing and prefabricated crack are assembled, as shown in Figure 12. In order to make the calculation results in crack propagation easier to converge, the loading process of the tubing is simplified as the cyclic reciprocating motion of the loaded end in the same direction. In the loop module, the tubing loaded section is coupled to its center RP-1. Then, it is constrained in boundary conditions. When the boundary condition constraints are imposed on the tubing, the six freedom degrees of 20 mm fixed end at the left end are all constrained. Point RP-1 is subjected to cyclic displacement. The cycle time displacement curve is shown in Figure 13, where *T* is the time used for one cycle of cyclic loading, with the amplitude of the displacement value of $\Delta X = 6$ mm.

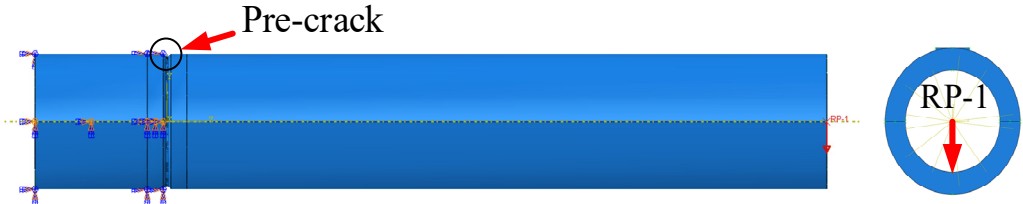

**Figure 12.** Application of boundary condition.

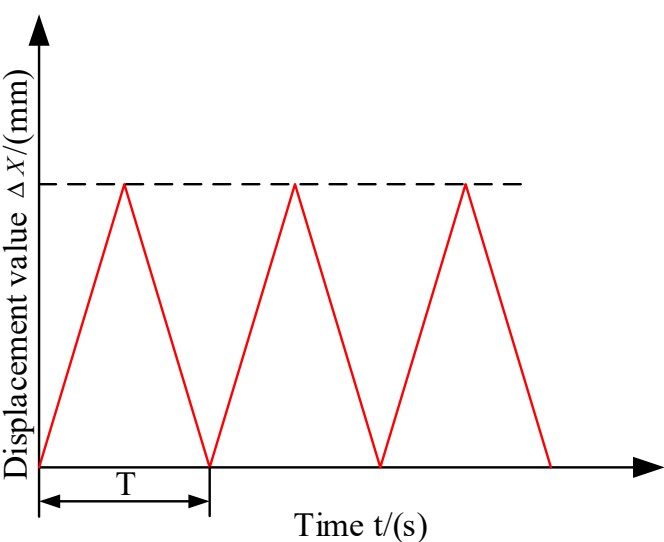

**Figure 13.** Tubing loading time displacement curve.

The 304 stainless steel tubing is selected as the research object. For detailed data, the elastic modulus is $2.1 \times 10^5$ MPa, the Poisson's ratio is 0.3, and the density is 7850 kg/m$^3$. In addition to material properties, the analysis of crack propagation requires additional parameters to be added to the keyword [19,20]. The maximum principal stress criterion is chosen as the failure criterion, and the energy-based mixed-mode exponential evolution law is chosen for the damage evolution. When dividing the tubing mesh, the area near the V-groove is densified, with a cell size of 0.5 mm × 0.5 mm × 0.25 mm. The size of the rest of the grid cells is 1.0 mm × 0.5 mm × 0.5 mm. Prefabricated crack does not require meshing.

*5.2. Crack Propagation Process of Tubing Fatigue Fracture Precision Separation*

Figure 14 shows the PHILSM cloud image of tubing outer surface fatigue crack propagation. According to the crack propagation trajectory, the crack mainly propagates forward along the section where the V-groove is located. In the early stage of fatigue crack propagation, the crack grows along the V-groove section. In addition, the crack propagation direction is shifted toward the loading end side. In the late stage of fatigue crack propagation, the crack propagates toward the bottom of the V-groove.

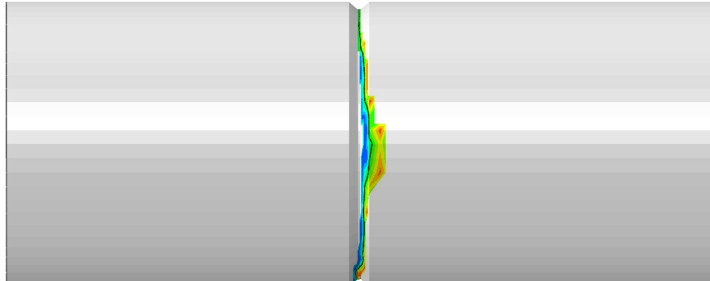

**Figure 14.** PHILSM cloud image of fatigue crack propagation.

Figure 15 shows the STATUSXFEM cloud image of the tubing fatigue fracture precision separation process. Among them, the value of STATUSXFEM is between 0 and 1. STATUSXFEM = 1 means complete break. STATUSXFEM = 0 means no cracking. STATUSXFEM between 0 and 1 indicates different degrees of fragmentation. The fatigue crack propagation is analyzed according to the crack surface propagation process. When loading cycles $N = 0$, prefabricated crack appears. When loading cycles $N$ is 10–100, the crack propagation is stable, and the section is smooth, namely, the stable propagation region. When loading cycles $N$ is 150–300, the crack propagation fluctuates slightly. Additionally, partially shaded

areas and bumps appear. Overall, the extension is relatively complete. When loading cycles *N* is 350–450, there is a large deviation in the crack propagation direction, which leads to the unsmooth local expansion, that is, the unsmooth propagation region. When loading cycles *N* is 500, the final fracture region occurs. In the later stages of crack propagation, tubing section separation is affected by crack propagation. Due to eccentric load as well as cantilever beam state, the tubing separation is mainly due to the tearing of material, and finally, the instantaneous break occurs. Therefore, the last separated section region does not have the characteristics of stable crack propagation. According to Figure 15o, 1 represents the stable propagation region, 2 represents the unsmooth propagation region, and 3 represents the final fracture region. It can be seen from the longitudinal crack surface in Figure 15p that the crack surface in the early stage of fatigue crack propagation is flat, while the crack surface in the later stage of crack propagation is unsmooth.

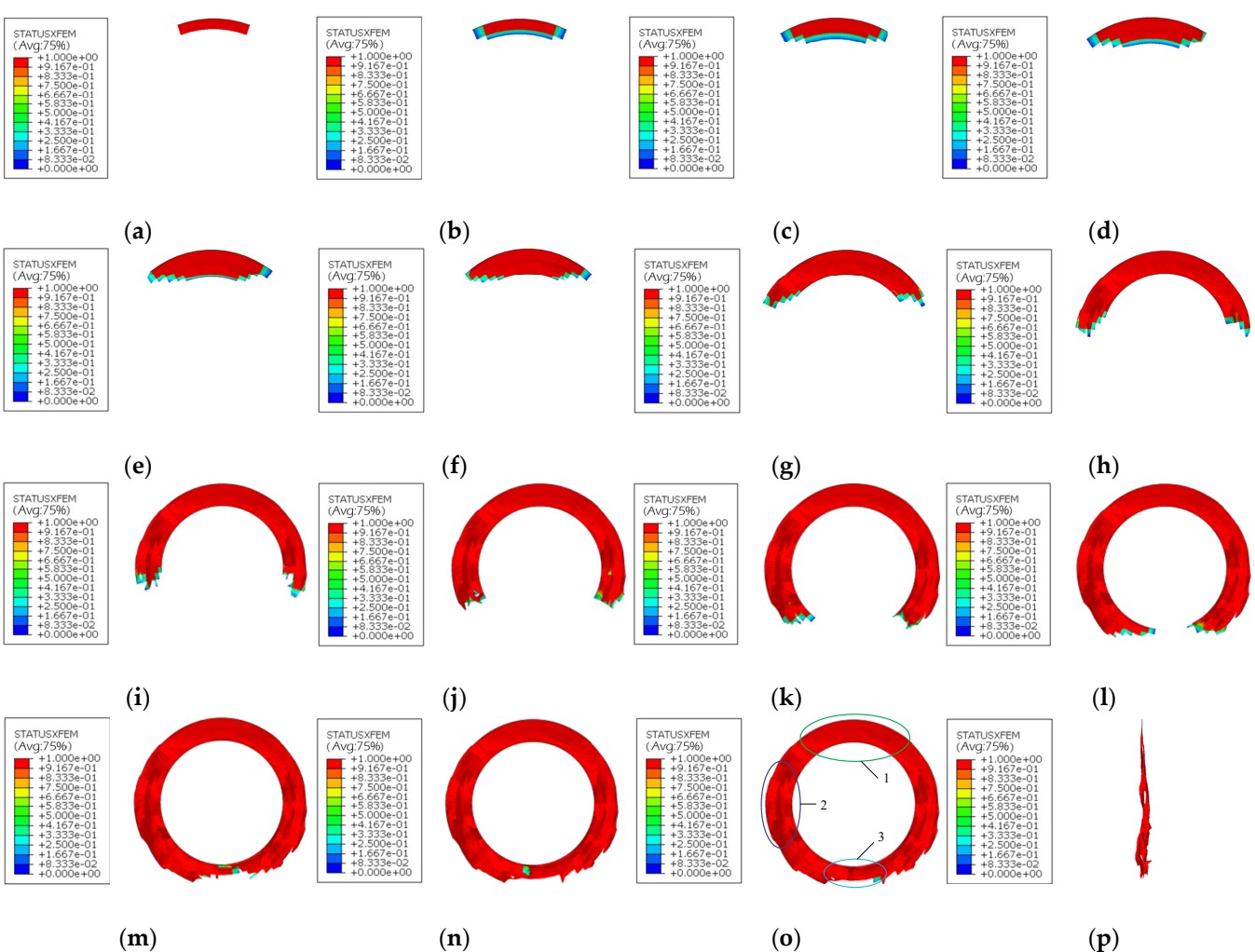

**Figure 15.** Tubing fatigue fracture precision separation process: (**a**) N = 0; (**b**) N = 10; (**c**) N = 20; (**d**) N = 30; (**e**) N = 40; (**f**) N = 50; (**g**) N = 100; (**h**) N = 150; (**i**) N = 200; (**j**) N = 250; (**k**) N = 300; (**l**) N = 350; (**m**) N = 400; (**n**) N = 450; (**o**) N = 500; (**p**) Crack face.

According to Figure 15, during the finite element propagation process of the crack in the tubing fatigue fracture precision separation, the propagation speed of the outer ring crack is greater than that of the inner ring. With the increase in loading cycles, the crack surface gradually changes from smooth to undulating, and the overall section is gradually uneven. In the crack propagation process, the time between the crack initiation stage and the instability transient stage is shorter than that in the steady-state propagation stage.

For the experimental setup, the same type of 304 stainless steel tubing as the simulation is chosen. Finally, the tubing separation section shown in Figure 16 is obtained. There are three main regions in the section, namely, stable propagation region, unsmooth propagation region and final fracture region. The positions of the three regions agree with the section positions obtained from the crack propagation process simulation of tubing fatigue fracture. Therefore, the tube fatigue crack propagation precision separation process is validated.

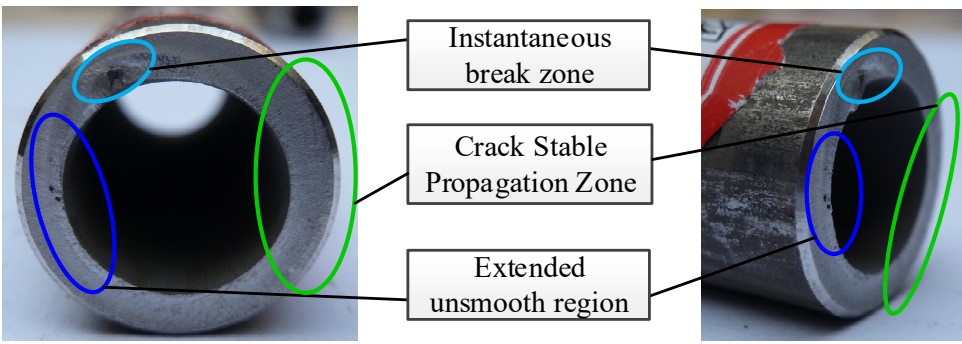

Instantaneous break zone

Crack Stable Propagation Zone

Extended unsmooth region

**Figure 16.** Tubing fatigue fracture precision separation section.

## 6. Conclusions

(1) Through the quasi-static tensile experiment, the mechanical properties of 304 stainless steel tubing section sample are studied. The true stress–true strain curve of 304 stainless steel tube was obtained. Through theoretical derivation and nonlinear fitting, the Johnson–Cook constitutive model of 304 stainless steel tube separation at room temperature was constructed, which laid a foundation for accurately simulating the precise separation process of tubes.

(2) The finite element model of eccentric loading was established to simulate the stress change of the tube under different rotation angles of the eccentric. During the rotation of the eccentric from $0°$ to $90°$, the stress concentration position of the V-groove section of the tube changes continuously, the values of the maximum axial tensile stress and compressive stress continue to increase, and in the process of expansion, the axial stress continues to gradually expand from the outer ring of the V-groove to the inner ring when the eccentric rotates from $135°$ to $225°$. During the process, the value of the maximum axial tensile stress of the tube does not change much, but the axial stress value of the inner ring of the V-groove section of the tube continues to increase. When the eccentric wheel rotates to $360°$, the axial direction of the V-groove section of the tube increases. The maximum axial tensile stress and maximum compressive stress of the V-groove section of the tube are at the same position. The axial stress changes at the symmetrical position of the root of the V-shaped groove of the tube are also symmetrical to each other. The greater the eccentricity, the greater the axial stress.

(3) The three-dimensional finite element dynamic simulation of crack propagation during fatigue fracture of 304 stainless steel tube is realized. When the tube is precisely separated, in the process of fatigue loading cycles from 0 to 500 cycles, the crack growth rate of the outer ring of the tube is greater than that of the inner ring. With the increase in the loading cycle, the section gradually changes from smooth to high and low, and finally, there is an obvious transient break area, which is consistent with the experimental results.

**Author Contributions:** Conceptualization and methodology, R.Z., R.P., W.G. and D.Z.; Formal analysis, R.Z. and X.X.; Supervision, R.Z. and P.Z.; Writing—original draft preparation, X.Z.; Writing—review and editing, R.P. All authors have read and agreed to the published version of the manuscript.

**Funding:** This research was supported by the China Postdoctoral Science Foundation (Fund number: 2018M633543).

**Institutional Review Board Statement:** Not applicable.

**Informed Consent Statement:** Not applicable.

**Data Availability Statement:** Not applicable.

**Conflicts of Interest:** The authors declare no conflict of interest.

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
