# Peer review of "Finite Element Analysis of Crack Propagation in Precise Separation Process of Tubing Eccentric Loading Fatigue Fracture"

_applsci, doi:10.3390/app12105111_

Round 1

Reviewer 1 Report

Manuscript Ref: applsci-1703206 

The authors presented a technique to study fatigue crack growth in tubular components with axi-symmetric notch. By using an offset for the centre of tube with respect to rotating axis, the strain was induced in the notched section. 

Following are the comments.

  1.    The title is very general and it doesn’t reflect the correct objective of the paper. It must be changes.
    2.    The abstract should be shortened and it should be modified by correctly specifying the objectives and outcomes of this research.
    3.    Fig. 1 is not clear. Why are there so many grooves in tube? Are you applying loads on clamping or is it fixed at one position? Please clarify and modify the figure accordingly.
    4.    Table 2 should be removed. The data is not used in the calculation.
    5.    Fig. 2 is misleading. The hatched cross-section is wrongly shown. It seems like there are two tubes. Please correct this figure.
    6.    Fig. 3 is not necessary. Please remove this.
    7.    Equation (1) is obvious and need not be mentioned. Just mention the rate of loading in the text.
    8.    Fig. 5 is not necessary. It is very standard and should be removed.
    9.    Fig. 7: Why are data shown upto 4% only? For SS304L, the strain at failure at room temperature is of the order of 80. Please show the whole stress-strain curve for all rates. Both engineering and true stress curves should be shown.
    10.    Fig. 7: The data doesn’t seem to be strain-rate sensitive. Please explain and elaborate.
    11.    Please show your fitting of curves for Johnson-Cook model. The method of finding the parameters should be elaborated.
    12.    It is not clear why J-C parameters are evaluated. Where are these used in this paper?
    13.    Fig. 9: Please mention the offset value between O1 and o2 in this figure. Elaborate the reason behind the choice of a particular value of offset.
    14.    Fig. 11: The contour are not clear. Please increase the size of legends.
    15.    Fig. 13: Only axial stresses are plotted. As the tube has a notch, this will induce stress multiaxiality. The variation of von Mises stress should be shown as a function of rotation angle.
    16.    Fig. 17: The legend are not legible in Fig. 17. Please improve the legends.
    17.    Section 5.2: The authors write about fatigue crack propagation. There is no data presented regarding fatigue crack growth rates (da/dN) with number of cycles. Please elaborate how is fatigue crack growth measures? Please plot the da/dN vs. Delta_K graph. Elaborate the method and results.
    18.    Conclusions: The authors write ‘the stress concentration effect accelerates the crack growth rate’. This is very obvious. The fatigue crack growth rate must be quantified. The stress multiaxiality near the notch should be quantified. How is the strain rate related to fatigue crack growth experiment? Please explain. The conclusion section should be modified and only conclusions relevant to this work should be mentioned.

The manuscript should be revised thoroughly by elaborating the experimental procedure, fatigue crack growth measurement, analysis of notched region stresses etc. and linking the measurement data with FE analysis results.

Reviewer 2 Report

1. Introduction should be improved to emphasize the importance of research.

2. What was mesh density? Cells numbers? Please precise

3. Quality of the Figure 11 is poor.

4. To improve reading and comparison, Figure 13 a-c could be presented as a one figure. The same for fig.13 b-d

5. page 9, 2nd paragraph–repeating

Question

Section 5. The authors used triangular time displacement curve (figure 15).

The course of axial stress over time is rectangular (fig 13). Have the authors performed research for a squared displacement-course? Such research could improve the paper.

Reviewer 3 Report

Dear author/s,

This paper is generally very well written and scientifically sound. However, I would suggest a couple of changes, which do not affect the structure of the paper:

Some spelling mistakes were made. Except standard spelling mistakes, authors are using United States English. Words that could be corrected are: mold/mould, analyzed/analysed, centerline/centreline.

Page 2 - “2. Principle of tubing prec...” - Experimental method could be explained better, since it is hard to understand the testing method completely.

Page 8 – Noticed some double-spaces, please check.

Page 12 – Conclusion is a bit anticlimactic. It seems that the research has scientific potential, but the conclusion is too basic and doesn’t seem to provide a major scientific contribution. For example, the paper is concluded with “...the stress

concentration effect accelerates the crack growth rate.” This is expected behaviour and already widely known. The conclusion should be improved by the authors for a journal of this status.

Best Regards!

Round 2

Reviewer 2 Report

Manuscript is improved according to my comments. Accept in present form